

# Holographic Weyl anomaly in string theory

Lorenz Eberhardt[1⋆] and Sridip Pal[1,2†]

**1** School of Natural Sciences, Institute for Advanced Study,
1 Einstein Drive, Princeton, NJ 08540, USA
**2** Walter Burke Institute for Theoretical Physics,
California Institute of Technology, Pasadena, CA, USA

⋆ elorenz@ias.edu , † sridip@caltech.edu

## Abstract

We compute the worldsheet sphere partition function of string theory on global AdS$_3$ with pure NS-NS flux. Because of an unfixed Möbius symmetry on the worldsheet, there is a cancellation of infinities and only a part of the answer is unambiguous. We show that it precisely reproduces the holographic Weyl anomaly and the ambiguous terms correspond to the possible counterterms of the boundary CFT.



# 1 Introduction

The seemingly simplest string theory diagrams are paradoxically under poorest conceptual control since they often reflect ambiguities of the theory. This happens when the worldsheet has a residual non-compact automorphism group. Since it is part of the gauge group in string theory, the infinite volume needs to be canceled by a corresponding target space volume divergence. The possible topologies with a non-compact automorphism group are the sphere with 0, 1 or 2 punctures and the disk with 0, 1 or 2 boundary punctures. The disk case and the two-point function on the sphere can be treated without ambiguities for example by imposing a suitable gauge-fixing condition [1–4]. However, this leaves out the arguably most interesting case – the sphere partition function.

The sphere partition function conjecturally captures the on-shell gravitational action of the background and as such carries the usual ambiguities [5]. Using standard worldsheet techniques, it has only been computed directly in some very special situations such as the minimal string [6] or in two-dimensional gravity [7]. For compact target spaces, the divergence from the infinite volume of the automorphism group is uncompensated and the sphere partition function vanishes [8]. It has also been proposed that it might be necessary to give up conformality on the worldsheet to compute the sphere partition function in general [9,10].

We explain the physically relevant example of $AdS_3$ backgrounds where the sphere partition function can be directly understood within the standard framework of string perturbation theory. Via the AdS/CFT correspondence, these string backgrounds are dual to two-dimensional CFTs [11–13]. The sphere partition function captures the conformal anomaly in the boundary CFT and reflects the process of holographic renormalization in string theory [14]. Contrary to ordinary gravity, we don't have to do any adhoc regularization of the gravitational on-shell action.[1]

The method of our computation is not new – we exploit that we can compute derivatives of the sphere partition function by inserting zero-momentum dilaton vertex operators which we explain in Section 2. The interpretation of the result is however interesting and we discuss it in detail in Section 3. For the actual computation, we need to carefully work out the normalization of the sphere partition function, which we do in Appendix A.

# 2 Computing the sphere partition function

We consider superstrings on $AdS_3 \times X$ with pure NS-NS flux. We are agnostic about the compactification $X$ since it will not play a role. We are only interested in genus 0 and thus do not have to specify the GSO projection. We merely have to assume that $X$ is a compact sigma model described by an $\mathcal{N} = 1$ sigma-model on the worldsheet of the correct central charge. The $AdS_3$ part of the background is described by the $SL(2,\mathbb{R})$ WZW model at level $k + 2$, together with three free fermions [16]. The most important part of the worldsheet theory is the $SL(2,\mathbb{R})$ WZW model whose relevant features we briefly recall (or more precisely we are actually considering an analytic continuation of the WZW model to Euclidean spacetime signature known as the $H_3^+$ model). The fermions do not influence our computations, except for shifting the level $k \to k + 2$ and setting the correct normalization for the dilaton vertex operator.

---

[1] [15] computed the one-point function on $AdS_3$ using a certain cutoff procedure. Hence it might also be possible to implement such a procedure for the sphere partition function itself.

## 2.1 The action

The action of the $SL(2,\mathbb{R})_{k+2}$ model reads[2]

$$S_{SL(2,\mathbb{R})} = \frac{k+2}{\pi} \int d^2z \left( \partial\phi\bar{\partial}\phi + e^{2\phi}\partial\bar{\gamma}\bar{\partial}\gamma \right). \tag{2.1}$$

This is simply the action for a sigma model on Euclidean global AdS$_3$ in Poincaré coordinates.[3] The metric in these coordinates reads

$$ds^2_{AdS} = d\phi^2 + e^{2\phi} d\gamma\, d\bar{\gamma}. \tag{2.2}$$

The boundary of AdS$_3$ is located at $\phi \to \infty$.[4] We use units in which $\alpha' = 1$ so that $k$ corresponds to the radius of AdS$_3$ measured in units of string length. The $B$-field $B = k\, d\gamma \wedge d\bar{\gamma}$ is responsible for the absence of the term $e^{2\phi}\partial\gamma\bar{\partial}\bar{\gamma}$ in the action. The shift $k \to k+2$ is a one-loop effect and thus we omitted it in semiclassical quantities.

It is convenient to modify the action slightly and introduce a coupling constant $\mu$ analogous to the cosmological constant in Liouville theory:

$$S_{SL(2,\mathbb{R})} = \frac{1}{4\pi} \int d^2z \left( 4(k+2)\partial\phi\bar{\partial}\phi + \mu e^{2\phi}\partial\bar{\gamma}\bar{\partial}\gamma \right). \tag{2.3}$$

This seems to be unimportant since we can always remove $\mu$ by redefining $\phi \to \phi - \frac{1}{2}\log\mu$. However, the correlation functions and partition functions will not be quite invariant under this shift since both the vertex operators and the path integral measure are not invariant. This leads to a version of the KPZ scaling argument [18]. The path integral measure takes the form

$$\mathscr{D}\phi\,\mathscr{D}(e^\phi\gamma)\,\mathscr{D}(e^\phi\bar{\gamma}), \tag{2.4}$$

which is induced from the Haar measure on $SL(2,\mathbb{R})$. One can remove the various factors of $e^\phi$ at the cost of introducing a Jacobian factor which corrects the action at one loop. The result is [16, 19–21]

$$S = \frac{1}{4\pi} \int d^2z \left( 4k\partial\phi\bar{\partial}\phi + R\phi + \mu e^{2\phi}\partial\bar{\gamma}\bar{\partial}\gamma \right), \tag{2.5}$$

where the path integral measure is now that of a free fields. This result can be derived in a variety of ways, for example by (i) by relating it to the chiral anomaly as in [19] and using the well-known formulas for the chiral anomaly in 2d, (ii) by using an index theorem counting the zero modes of $\gamma$ (and $\beta$ that is introduced to pass to a first order formalism), or (iii) by simply writing down the most general local expression of the correct dimension in $\phi$ and the metric, requiring the Jacobian to behave group-like under repeated removal of factors $e^\phi$ and matching the central charge to the known value of $SL(2,\mathbb{R})_{k+2}$.

One can see from (2.5) that the path integral over $e^{-S}$ converges for surfaces of negative Euler characteristics. Indeed, in this case the path integral is damped for $\phi \to +\infty$ thanks to the presence of the term $\mu e^{2\phi}\partial\bar{\gamma}\bar{\partial}\gamma$ and for $\phi \to -\infty$ thanks to the presence of the Ricci scalar.[5]

---

[2]Since numerical factors will be important, we notice that we define $d^2z = dx\,dy$ and $\partial = \frac{1}{2}(\partial_x - i\partial_y)$ for $z = x + iy$. We follow the conventions for sigma-models given e.g. in Polchinski [17, Section 3.7] with $\alpha' = 1$.

[3]We take $\bar{\gamma}$ to be the complex conjugate of $\gamma$. For Lorentzian AdS$_3$, they would both be real and independent. Thus we are considering an analytic continuation of the $SL(2,\mathbb{R})$ WZW model. As is common in the literature, we will nonetheless continue to refer to this CFT as the $SL(2,\mathbb{R})$ WZW model.

[4]The change of coordinates $z = e^{-\phi}$ gives a perhaps more standard form of the Poincaré metric.

[5]The suppression for $\phi \to -\infty$ can fail on a genus 0 surface or when including certain vertex operators. This leads to singularities in the correlators as a function of the spins. The suppression for $\phi \to +\infty$ can also fail because there might be a holomorphic map $\gamma$ with $\bar{\partial}\gamma = 0$. This leads to certain well-studied singularities in the correlation functions of the model as a function of the moduli [2, 22–24].

## 2.2 KPZ scaling and the coupling constant

From the form (2.5), it is simple to derive the $\mu$-dependence of a partition function. We can shift $\phi \to \phi - \frac{1}{2}\log\mu$. Only the term involving the Ricci scalar is not fully invariant under this shift and shows that the partition function has the $\mu$-dependence $\mu^{\frac{1}{2}\chi_g} = \mu^{1-g}$ as a consequence of the Gauss-Bonnet theorem. More generally, in the presence of $n$ vertex operators of $\mathrm{SL}(2,\mathbb{R})$ spin $j_i$, we get the following scaling of a correlation function:

$$\left\langle \prod_i V_{j_i} \right\rangle \propto \mu^{1-g-\sum_i j_i}. \tag{2.6}$$

This is the analogue of the KPZ scaling of Liouville theory [18]. We should note that this equation only follows from the path integral if the exponent of $\mu$ is negative, since otherwise the path integral does not converge. We will indeed see below that (2.6) can get modified for positive exponents of $\mu$.

We also note that $\mu$ plays the role of the string coupling in the theory, more precisely, we can identify

$$\mu \propto \frac{1}{g_s^2}. \tag{2.7}$$

In particular, there is no need to introduce a separate string coupling since it is already contained naturally in the $\mathrm{SL}(2,\mathbb{R})_{k+2}$ WZW model. In particular, weak string coupling corresponds to large values of $\mu$.

## 2.3 The sphere partition function and its derivatives

We are interested in computing the sphere partition function of superstrings on $\mathrm{AdS}_3 \times X$. As usual, this computation is subtle since one has to divide by the volume of the super Möbius group $\mathrm{OSP}(1|2,\mathbb{C})/\mathbb{Z}_2$, which is badly divergent [1]. Instead, one can use that insertion of the vertex operator

$$I = -\frac{1}{4\pi}\,\mathrm{e}^{2\phi}\partial\bar{\gamma}\bar{\partial}\gamma\,, \tag{2.8}$$

implements $\mu$-derivatives of correlation functions. One can read off from the exponent that it indeed has spin $j = 1$ consistent with the scaling (2.6). This is the zero mode of the dilaton vertex operator and has played a prominent role in $\mathrm{AdS}_3$ holography [13, 16, 25, 26]. From a path integral point of view we have for the $\mathrm{SL}(2,\mathbb{R})_{k+2}$ WZW model

$$\partial_\mu \left\langle \prod_i V_{j_i}(x_i,z_i) \right\rangle_g = \int \mathrm{d}^2 z \left\langle I(z) \prod_i V_{j_i}(x_i,z_i) \right\rangle_g, \tag{2.9}$$

since the dilaton vertex operator is precisely the marginal operator appearing with the coupling $\mu$ in eq. (2.5). Here we use the natural primary vertex operators $V_{j_i}(x_i,z_i)$ of the sigma-model on Euclidean $\mathrm{AdS}_3$. We note that the coordinate $x$ corresponds to the location of the vertex operator on the boundary of Euclidean $\mathrm{AdS}_3$. In view of the scaling (2.6), the left-hand side of (2.9) is simply the correlation function itself times the exponent $1 - g - \sum_i j_i$ and times $\mu^{-1}$. This equation can also be established directly from an axiomatic approach to the $\mathrm{SL}(2,\mathbb{R})$ WZW model [26, 27].

On the level of the integrated string theory correlators, this means that

$$\partial_\mu \left\langle\!\!\left\langle \prod_i V_{j_i}(x_i) \right\rangle\!\!\right\rangle_g = \left\langle\!\!\left\langle I \prod_i V_{j_i}(x_i) \right\rangle\!\!\right\rangle_g, \tag{2.10}$$

where the double brackets denote the integrated string theory correlators, which no longer depend on $z_i$.[6] Since we are working within the RNS formalism of superstring theory, we should also include picture changing operators or integrate over supermoduli space. We use the former formalism, but suppress this from our notation in the main text, where we just explain the bosonic formalism. We include the details in the computation of Appendix A.

The starting point for the sphere partition function is not a well-defined one, but it is reasonable to assume that it can be defined by assuming (2.10) to continue to hold. In particular, we learn that for the string theory sphere partition function $Z_{S^2}$ (which could also be represented by an empty double bracket), we can compute the third derivative without problems:

$$\partial_\mu^3 Z_{S^2} = \langle\langle I\,I\,I\rangle\rangle = C_{S^2}\langle I(0)I(1)I(\infty)\rangle\,, \tag{2.11}$$

since a string theory three-point function essentially coincides with the worldsheet three-point function up to ghosts and the normalization $C_{S^2}$ of the string theory path integral. The right hand side is a well-defined quantity in the $SL(2,\mathbb{R})$ WZW model and can be computed. It takes the form

$$\partial_\mu^3 Z_{S^2} = \langle\langle I\,I\,I\rangle\rangle = -F(k)\mu^{-2}\,, \tag{2.12}$$

for some function $F(k)$, which we compute in Appendix A. We used the KPZ scaling (2.6) for the $\mu$ dependence. We can thus integrate this equation back up to obtain the sphere partition function, up to three integration constants.

We could have done slightly better. It is known how to define the two-point function in string theory, see [2] for the case of $AdS_3$ and [3] for a more general discussion. Thus one can reliably compute the second derivative of $Z_{S^2}$ and obtains

$$\partial_\mu^2 Z_{S^2} = \langle\langle I\,I\rangle\rangle = F(k)\mu^{-1}\,. \tag{2.13}$$

This means that there is no integration constant when integrating (2.12) back up to (2.13). We thus obtain

$$Z_{S^2} = F(k)\mu\log\mu + C_1 + C_2\mu\,, \tag{2.14}$$

for two integration constants $C_1$ and $C_2$. We notice that they should be included since there is no natural scale in the logarithm and thus the first well-defined term can mix with the integration constants.

## 3 Interpreting the result

We now interpret the result (2.14) physically.

### 3.1 Expectation

Let us first review what result is expected. From a holography point of view, our computation should reproduce the large $N$ partition function of the dual CFT on the boundary of global Euclidean $AdS_3$, which also happens to be a two-sphere (not to be confused with the worldsheet two-sphere that we discussed above). The sphere partition function of a CFT is fully determined from the conformal anomaly. Assuming that the metric is a round sphere of radius $R$, it takes the form

$$\log Z_{CFT} = \frac{c}{3}\log R + N_1 + N_2 R^2\,, \tag{3.1}$$

---

[6]For arbitrary choices of spins $j_i$, one may need to combine the vertex operators with vertex operator from the internal CFT $X$ so that they satisfy the mass-shell condition. This won't be important for us.

where $c$ is the central charge. The constants $N_1$ and $N_2$ correspond to the two possible counterterms we can add to the theory. $N_1$ and $N_2$ arise since we can add the counterterms

$$\int \mathrm{d}^2 x \, \sqrt{g} R \quad \text{and} \quad \int \mathrm{d}^2 x \, \sqrt{g} \,, \tag{3.2}$$

to the action, which modifies the partition function accordingly. For a nice general discussion about these ambiguities, see e.g. [28]. Of course, it is well-known how to reproduce this structure from classical gravity via holographic renormalization [14, 29].

The sphere partition function in string theory should hence precisely compute this universal dependence of the partition function on the radius and reflect the possible ambiguities of the sphere partition function of the dual CFT. More concretely, we should have

$$Z_{S^2} + \mathcal{O}(\log \mu) \overset{!}{=} \log Z_{\text{CFT}} \,. \tag{3.3}$$

The logarithm on the right hand side appears since the full string partition function also receives contributions from several disconnected spheres. The $\log \mu$ corrections on the string side originate from the torus diagram, see Section 3.3 below.

## 3.2 Relating the coupling $\mu$ to the radius

Going back to the string theory calculation, we notice that the expected result (3.1) features an explicit dependence on the radius of the boundary sphere on which the dual CFT is defined, which of course is not a parameter that entered the string theory calculation.

From the path integral definition (2.3) of the $SL(2, \mathbb{R})$ action one can however plausibly relate the coupling $\mu$ to the size of the asymptotic boundary as follows. In the gravity calculation, we would put the AdS boundary explicitly on some profile $\phi = \phi(\gamma, \bar{\gamma})$ and $\phi$ would become the Weyl factor of the induced metric. With the inclusion of the coupling constant $\mu$, the induced metric is instead

$$\mu \, \mathrm{e}^{2\phi} \, \mathrm{d}\gamma \, \mathrm{d}\bar{\gamma} \,. \tag{3.4}$$

In particular, we see that $\mu$ is related to the size of the metric on the holographic surface. We can thus identify

$$\mu \propto R^2 \,, \tag{3.5}$$

where $R$ is the characteristic radius of the holographic surface.

Hence we see that we should interpret some of the $\mu$'s in (2.14) in terms of the radius of the holographic surface. We write

$$Z_{S^2} = 2F(k)\mu \log R + C_1 + C_2' \mu + C_2'' R^2 \,. \tag{3.6}$$

Intuitively, we should trade the non-analytic dependence on $\mu$'s that does not follow the expected naive scaling (2.6) for $R$'s. For the term proportional to $\mu$, it does not matter whether we interpret $\mu$ as $R^2$ and we included both terms. In any case, this has the correct form to match with (3.1), where we keep in mind that e.g. the central charge scales like $\mu \propto \frac{1}{g_s^2} \propto G_N^{-1}$ [30].

Besides having the correct form, the only meaningful comparison arises from comparing the precise coefficient of $\log R$. To claim success, we hence have to check that

$$6F(k)\mu = c \,, \tag{3.7}$$

reproduces the expected central charge $c$ of the boundary CFT.

Let us outline the strategy for this. In order to relate $F(k)$ to the central charge of the boundary, we also compute the two- and three-point function of the holographic stress tensor $T$ and assume it to have the correct form as expected by holography. This allows us to relate

the normalization of $I$ to the normalization of $T$, which in turn is related to the central charge. After computing

$$\langle\!\langle I\,I \rangle\!\rangle\,, \qquad \langle\!\langle I\,I\,I \rangle\!\rangle\,, \qquad \langle\!\langle T\,T \rangle\!\rangle \quad \text{and} \quad \langle\!\langle T\,T\,T \rangle\!\rangle\,, \tag{3.8}$$

one can eliminate all normalizations and compute $F(k)$ unambiguously. This then confirms eq. (3.7). The reader can find the details of the computation in Appendix A.

### 3.3 A note on the torus partition function

Let us briefly comment on the torus partition function of global AdS$_3$. It suffers from a milder version of the same problem as the sphere, since the worldsheet path integral does not converge on a surface of genus 1. Indeed, the Ricci term in (2.5) is missing and thus the path integral is unsuppressed in the region $\phi \to -\infty$. Instead, we can compute the one-point function on the torus of the vertex operator $I$ defined in eq. (2.8) to compute the $\mu$-derivative. Integrating with respect to $\mu$, then predicts that the string theory torus partition function takes the form

$$Z_{\mathbb{T}^2} = G(k)\log\mu + C_3 \sim 2G(k)\log R + C_3\,, \tag{3.9}$$

where we again reinterpret $\mu$ in terms of the size of the boundary surface as in (3.5). In particular, the torus partition function can compute a possible one-loop correction to the central charge. Contrary to the sphere partition function, this now involves also the explicit form of the torus partition function of the internal CFT $X$ and thus the function $G(k)$ is background dependent. For example, in the background AdS$_3 \times$ S$^3 \times$ K3, it is known that the one-loop correction to the central charge is $+6$ and hence $G(k) = 1$, see e.g. [31] for a direct supergravity computation. We are however not aware of a corresponding worldsheet computation to confirm this.

We learn in particular also that the central charge of the dual CFT is always one-loop exact since higher genus partition functions will never contain logarithmic contributions.

## 4 Discussion

Let us mention a few open questions.

**Bosonic string.** One can repeat the same computation described in this paper for the bosonic string, but in this case the relation (3.7) is not satisfied. Instead, one finds

$$F_{\text{bos}}(k) = \frac{c_{\text{bos}}(k-2)}{24k\mu}\,, \tag{4.1}$$

$c_{\text{bos}}$ is again the boundary central charge in this case (as appearing in the OPE of the boundary stress tensor). The fact that this does not match the expectation is perhaps not too worrying since there is no well-defined boundary CFT in this case because of the tachyon. We find it nevertheless puzzling and don't have a good explanation for the numerical value.

**General backgrounds.** One may wonder whether this method of computation generalizes to other backgrounds. To a certain degree, the answer is yes – we believe that one can compute all the non-analytic terms in the sphere partition function using this method in an arbitrary background. In a given background one has to identify the zero-momentum dilaton vertex operator and insertion of this operator leads to suitable derivatives of the sphere partition function. It would be very desirable to carry out this program and confirm that the sphere partition function indeed agrees with the on-shell action as computed in gravity.

However, sometimes the sphere partition function is expected to be analytic, but still well-defined, in which case this method can fail. An example within string theory is the background of a stack of NS5-branes. The transverse directions can be described by the $\mathcal{N} = 4$ cigar [32,33], which is closely related to a coset of the SL(2, $\mathbb{R}$) WZW model discussed in this paper.[7] In this case it is reasonable to suspect that the sphere partition function computes (minus) the tension of the stack of NS5-branes, which is indeed proportional to $g_{\mathrm{s}}^{-2}$ (as defined in flat space), but does not contain any logarithmic terms. As a consequence, the second derivative of the sphere partition function is actually zero and thus the tension is hidden in one of the integration constants. Such situations also arise in the computation of black hole entropies. For example, the Schwarzschild black hole can be $\alpha'$ corrected and embedded in string theory. The string theory sphere partition function should compute the free energy, which is fully analytic. In such cases, there does not seem to be any known way to compute the sphere partition function directly in string theory without reducing it to a supergravity computation.

# Acknowledgments

We would like to thank Amr Ahmadain, Matt Heydeman, Shota Komatsu, Adam Levine, Raghu Mahajan, Juan Maldacena and Kostas Skenderis for discussions and Andrea Dei for comments on an early draft. We would like to thank one of the referees for pointing out a crucial sign error in the first version.

**Funding information** LE is supported by the grant DE-SC0009988 from the U.S. Department of Energy. SP acknowledges the support by the U.S. Department of Energy, Office of Science, Office of High Energy Physics, under Award Number DE-SC0011632 and by the Walter Burke Institute for Theoretical Physics.

# A  Fixing the normalization

In this appendix, we fix the precise function $F(k)$ in (2.14) for the RNS superstring.

## A.1  Picture numbers

In the main text we neglected picture numbers of the RNS superstring; or fermionic moduli space integrations in the language of supermoduli space [34]. In the superstring, the vertex operator (2.8) sits in a supermultiplet

$$I = I^{(-1,-1)} + \theta I^{(0,-1)} + \bar\theta I^{(-1,0)} + \theta\bar\theta I^{(0,0)}, \tag{A.1}$$

where $\theta$ is a Grassmann variable. Here the superscripts refer to the picture numbers of the vertex operator. We have

$$I^{(0,0)} = G_{-\frac{1}{2}}\bar{G}_{-\frac{1}{2}}I^{(-1,-1)}, \tag{A.2}$$

where $G_{-\frac{1}{2}}$ is the $\mathcal{N} = 1$ supercharge on the worldsheet. OSP(1|2, $\mathbb{C}$)/$\mathbb{Z}_2$ has two complex fermionic directions which means that the picture number of a correlation function has to sum up to $-2$ in a correlation function. Thus in a two-point function we can choose both vertex operators to have picture number $-1$, while we need to picture raise one of the vertex operators in a three-point function. Picture raising is equivalent to integration over a single

---

[7]For this, the NS5 branes have to be arranged in a particular circular configuration.

supermodulus [34]. It also involves additional terms with ghosts, but they do not contribute to correlation functions on the sphere. See e.g. [35, Chapter 13.2] for a standard discussion.

Concretely, this means that (2.10) reads more precisely

$$\partial_\mu^3 Z_{S^2} = C_{S^2} \langle I^{(-1)}(0) I^{(-1)}(1) I^{(0)}(\infty) \rangle, \tag{A.3}$$

where we wrote the common left and right picture number only once. Of course we can choose the normalization of picture raising in an arbitrary way, but we will compare the normalization to the correlation function of the stress tensor, and all these normalization factors will cancel out.

## A.2 Current algebra and vertex operators

The analytically continued $SL(2, \mathbb{R})_{k+2}$ WZW model describing string theory on Euclidean AdS$_3$ has primary vertex operators $V_j(x, z)$. Even though they are of course not holomorphic, we suppress right-moving coordinates. There are also so-called spectrally flowed vertex operators, which however we do not need for our discussion [2, 36, 37]. The spin $j$ can take either values in $\frac{1}{2} + i\mathbb{R}$ corresponding to the principal series representation of $\mathfrak{sl}(2, \mathbb{R})$ or in the interval $\frac{1}{2} < j < \frac{k+1}{2}$ corresponding to the discrete series representation. The latter correspond to short string states in target space [38]. All states of interest to us are of the latter type.

The primary fields $V_j(x, z)$ can be defined by translating with the operators $J_0^+$ and $J_0^-$ which correspond to the translation operators on the boundary of Euclidean AdS$_3$,

$$V_j(x, z) := e^{x J_0^+ + \bar{x} \bar{J}_0^+} V_j(0, z) e^{-\bar{x} \bar{J}_0^+ - x J_0^+}. \tag{A.4}$$

They have the following OPE with currents

$$J^+(\zeta) V_j(x, z) \sim \frac{1}{\zeta - z} \partial_x V_j(x, z), \tag{A.5a}$$

$$J^3(\zeta) V_j(x, z) \sim \frac{1}{\zeta - z} (x \partial_x + j) V_j(x, z), \tag{A.5b}$$

$$J^-(\zeta) V_j(x, z) \sim \frac{1}{\zeta - z} (x^2 \partial_x + 2x j) V_j(x, z). \tag{A.5c}$$

The OPE with $\bar{J}^a$ is analogous.

The modes of the currents and the adjoint fermions satisfy the (anti)commutation relations

$$[J_m^3, J_n^\pm] = \pm J_{m+n}^\pm, \qquad [J_m^+, J_n^-] = km \delta_{m+n} - 2 J_{m+n}^3, \qquad [J_m^3, J_n^3] = -\frac{km}{2} \delta_{m+n}, \tag{A.6a}$$

$$[J_m^3, \psi_r^\pm] = \pm \psi_{m+r}^\pm, \qquad [J_m^\pm, \psi_r^\mp] = \mp 2 \psi_{m+r}^3, \qquad [J_m^\pm, \psi_r^3] = \mp \psi_{m+r}^\pm, \tag{A.6b}$$

$$\{\psi_r^3, \psi_s^3\} = -\frac{k}{2} \delta_{r+s}, \qquad \{\psi_r^+, \psi_s^-\} = k \delta_{r+s}. \tag{A.6c}$$

We consider the NS sector and thus $r, s \in \mathbb{Z} + \frac{1}{2}$. It is also useful to define the decoupled currents

$$\mathcal{J}_m^\pm := J_m^\pm \pm \frac{2}{k} (\psi^3 \psi^\pm)_m, \qquad \mathcal{J}_m^3 := J_m^3 + \frac{1}{k} (\psi^- \psi^+)_m, \tag{A.7}$$

which satisfy a current algebra at level $\mathfrak{sl}(2, \mathbb{R})_{k+2}$ and commute with the fermions. Thus they are the currents of the $SL(2, \mathbb{R})_{k+2}$ WZW model that we discussed in the main text. The stress tensor and the supercharge, defining the $\mathcal{N} = 1$ Virasoro algebra on the worldsheet, are then given by

$$T = \frac{1}{2k} \left( \mathcal{J}^+ \mathcal{J}^- + \mathcal{J}^- \mathcal{J}^+ - 2 \mathcal{J}^3 \mathcal{J}^3 - \psi^+ \partial \psi^- - \psi^- \partial \psi^+ + 2 \psi^3 \partial \psi^3 \right), \tag{A.8a}$$

$$G = \frac{1}{k} \left( \mathcal{J}^+ \psi^- + \mathcal{J}^- \psi^+ - 2 \mathcal{J}^3 \psi^3 - \frac{2}{k} \psi^3 \psi^+ \psi^- \right), \tag{A.8b}$$

where normal ordering is implied. Of course, these also need to be combined with the internal CFT of the AdS$_3$ compactification in order to define a critical string worldsheet theory.

### A.3 Correlation functions

As discussed around eq. (2.8), we are interested in correlators of the dilaton vertex operator, which is a descendant of the spin $j = 1$ vertex operator. Thus, we will need the three point function of $V_1$, which is given by

$$\Big\langle \prod_{i=1}^{3} V_1(x_i, z_i) \Big\rangle = \frac{f_{VVV}\mu^{-2}}{|x_1 - x_2|^2 |x_2 - x_3|^2 |x_3 - x_1|^2} \, . \tag{A.9}$$

The structure constant $f_{VVV}$ is known explicitly, but we do not require the precise form [36]. We have only spelled out its $\mu$-dependence following from (2.6) explicitly, since this will be important for us. The correlation function is $z_i$-independent since the conformal weight of $V_1$ is zero.

The vertex operator $I^{(-1)}$ is the unique vertex operator in the theory with vanishing $J_0^3$ eigenvalue. It takes the form

$$I^{(-1)} := N_I \, \psi_{-1/2}^- \bar{\psi}_{-1/2}^- V_1 \, , \qquad I^{(0)} := G_{-1/2} \bar{G}_{-1/2} I^{(-1)} = N_I \, \mathcal{J}_{-1}^- \bar{\mathcal{J}}_{-1}^- V_1 \, , \tag{A.10}$$

where the second equation is a straightforward calculation using the worldsheet supercharge (A.8b) as well as the commutation relations (A.6). We included an arbitrary normalization $N_I$. This is the more axiomatic way of writing the vertex operator (2.8).

It is then simple to compute the string theory two-point function. The two-point function of $V_1$ with $V_1$ involves a $\delta(0)$ and is thus divergent. This is compensated in string theory by the division of the volume of the residual Möbius symmetry and one obtains a finite result.[8] We thus conclude that (see eq. (2.13) for the first equality)

$$F(k)\mu^{-1} = \langle\!\langle I^{(-1)}(x_1) I^{(-1)}(x_2) \rangle\!\rangle = k^2 C_{S^2} N_I^2 f_{VV} \mu^{-1} \, , \tag{A.11}$$

where $f_{VV}$ is the two-point function of $V_1$ with $\delta(0)$ stripped off, which is cancelled by the Möbius volume. We also included the (super)ghosts, which cancel the $z_i$-dependence of the correlation function. We again spelled out the $\mu$-dependence explicitly. The factor $k^2$ is important, it came from removing the left- and right-moving fermions, which leads to the factor of $k$ appearing in the anticommutator $\{\psi_r^+, \psi_s^-\}$, see eq. (A.6).

Let us illustrate this computation a little more in detail for the case of the three point function $\langle\!\langle I^{(-1)}(x_1) I^{(-1)}(x_2) I^{(0)}(x_3) \rangle\!\rangle$ of $I$. We begin by evaluating the corresponding CFT 3-point correlator for these operators. We can remove the free fermion and current algebra modes as follows:

$$\big\langle I^{(-1)}(x_1, z_1) I^{(-1)}(x_2, z_2) I^{(0)}(x_3, z_3) \big\rangle$$

$$= N_I^3 \oint_{z_1} \frac{\mathrm{d}z}{z - z_1} \Big\langle \big(\psi^-(z) - 2x_1 \psi^3(z) + x_1^2 \psi^+(z)\big)\big(\bar{\psi}_{-1/2}^- V_1\big)(x_1, z_1)$$

$$\times \big(\psi_{-1/2}^- \bar{\psi}_{-1/2}^- V_1\big)(x_2, z_2)\big(\mathcal{J}_{-1}^- \bar{\mathcal{J}}_{-1}^- V_1\big)(x_3, z_3) \Big\rangle \tag{A.12}$$

$$= \frac{N_I^3 k(x_1 - x_2)^2}{z_1 - z_2} \Big\langle \big(\bar{\psi}_{-1/2}^- V_1\big)(x_1, z_1)\big(\bar{\psi}_{-1/2}^- V_1\big)(x_2, z_2)\big(\mathcal{J}_{-1}^- \bar{\mathcal{J}}_{-1}^- V_1\big)(x_3, z_3) \Big\rangle \tag{A.13}$$

$$= \frac{N_I^3 k(x_1 - x_2)^2}{z_1 - z_2} \oint_{z_3} \frac{\mathrm{d}z}{z - z_3} \Big\langle \big(\bar{\psi}_{-1/2}^- V_1\big)(x_1, z_1)\big(\bar{\psi}_{-1/2}^- V_1\big)(x_2, z_2)$$

$$\times \big(\mathcal{J}^-(z) - 2x_3 \mathcal{J}^3(z) + x_3^2 \mathcal{J}^+(z)\big)\big(\bar{\mathcal{J}}_{-1}^- V_1\big)(x_3, z_3) \Big\rangle \tag{A.14}$$

---

[8]There is a finite factor remaining in this cancellation, but it is universal and thus unimportant for us [2, 25].

$$= -\frac{N_I^3 k (x_1 - x_2)^2}{z_1 - z_2} \sum_{i=1,2} \frac{1}{z_i - z_3} \left[ 2(x_i - x_3) + (x_i - x_3)^2 \partial_{x_i} \right]$$
$$\times \left\langle (\bar{\psi}_{-1/2}^- V_1)(x_1, z_1)(\bar{\psi}_{-1/2}^- V_1)(x_2, z_2)(\bar{\mathcal{J}}_{-1}^- V_1)(x_3, z_3) \right\rangle \qquad \text{(A.15)}$$

$$= \frac{N_I^3 k (x_1 - x_2)(x_2 - x_3)(x_3 - x_1)}{(z_1 - z_2)(z_1 - z_3)} \left\langle (\bar{\psi}_{-1/2}^- V_1)(x_1, z_1)(\bar{\psi}_{-1/2}^- V_1)(x_2, z_2)(\bar{\mathcal{J}}_{-1}^- V_1)(x_3, z_3) \right\rangle, \qquad \text{(A.16)}$$

where we used the $x$-dependence of the three-point function (A.9) in the last step. Now we repeat the same process to strip off the antiholomorphic modes and obtain

$$\langle I^{(-1)}(x_1, z_1) I^{(-1)}(x_2, z_2) I^{(0)}(x_3, z_3) \rangle = \frac{N_I^3 k^2 |x_1 - x_2|^2 |x_2 - x_3|^2 |x_3 - x_1|^2}{|z_1 - z_2|^2 |z_1 - z_3|^2} \left\langle \prod_{i=1}^{3} V_1(x_i, z_i) \right\rangle \qquad \text{(A.17)}$$

$$= \frac{N_I^3 k^2 f_{VVV} \mu^{-2}}{|z_1 - z_2|^2 |z_2 - z_3|^2}. \qquad \text{(A.18)}$$

The superconformal ghosts cancel the $z$-dependence of this correlator and we obtain for the string theory correlator

$$\langle\langle I^{(-1)}(x_1) I^{(-1)}(x_2) I^{(0)}(x_3) \rangle\rangle = k^2 N_I^3 C_{S^2} f_{VVV} \mu^{-2}. \qquad \text{(A.19)}$$

We have, see (2.13) and (2.12)

$$\partial_\mu \langle\langle I^{(-1)}(x_1) I^{(-1)}(x_2) \rangle\rangle = \langle\langle I^{(-1)}(x_1) I^{(-1)}(x_2) I^{(0)}(x_3) \rangle\rangle, \qquad \text{(A.20)}$$

which gives

$$N_I f_{VVV} = -f_{VV}. \qquad \text{(A.21)}$$

This of course does not completely determine the function $F(k)$ appearing in (2.13) and (2.12). We will fix it by computing the correlation function of the stress energy tensor on the boundary. The operator corresponding to the stress tensor on the boundary is the unique physical operator with $J_0^3$ eigenvalue 2 and $\bar{J}_0^3$ eigenvalue 0 (and is thus a holomorphic field of conformal weight $(2, 0)$ on the boundary). It takes the form

$$T^{(-1)} := N_T \left( \psi_{-1/2}^+ - J_0^+ \psi_{-1/2}^3 + \frac{1}{2}(J_0^+)^2 \psi_{-1/2}^- \right) \bar{\psi}_{-1/2}^- V_1 \qquad \text{(A.22)}$$

$$= 3 N_T \left( \psi_{-1/2}^+ - \psi_{-1/2}^3 J_0^+ + \frac{1}{2} \psi_{-1/2}^- (J_0^+)^2 \right) \bar{\psi}_{-1/2}^- V_1. \qquad \text{(A.23)}$$

The corresponding operator in picture 0 takes the form

$$T^{(0)} := G_{-1/2} T^{(-1)} = N_T \left( J_{-1}^+ - J_0^+ J_{-1}^3 + \frac{1}{2}(J_0^+)^2 J_{-1}^- \right) \bar{\mathcal{J}}_{-1}^- V_1. \qquad \text{(A.24)}$$

We compute

$$\langle\langle T^{(-1)}(x_1) T^{(-1)}(x_2) \rangle\rangle = \frac{3 k^2 N_T^2 C_{S^2} f_{VV} \mu^{-1}}{(x_1 - x_2)^4}, \qquad \text{(A.25)}$$

where we again allowed for an arbitrary normalization for the vertex operator $T$. From the perspective of the dual CFT with central charge $c$, we have

$$\langle\langle T^{(-1)}(x_1) T^{(-1)}(x_2) \rangle\rangle = \frac{c}{2(x_1 - x_2)^4}. \qquad \text{(A.26)}$$

Comparison of eqs. (A.25) and (A.26) tyields

$$6k^2 N_T^2 C_{S^2} f_{VV} \mu^{-1} = c \,. \tag{A.27}$$

We finally compute

$$\langle\langle T^{(-1)}(x_1) \, T^{(-1)}(x_2) \, T^{(0)}(x_3) \rangle\rangle = \frac{6k^2 N_T^3 C_{S^2} f_{VVV} \mu^{-2}}{(x_1 - x_2)^2 (x_2 - x_3)^2 (x_3 - x_1)^2} \,. \tag{A.28}$$

Comparing with the generic three-point function of the stress tensor in a CFT,

$$\langle\langle T^{(-1)}(x_1) \, T^{(-1)}(x_2) \, T^{(0)}(x_3) \rangle\rangle = \frac{c}{(x_1 - x_2)^2 (x_2 - x_3)^2 (x_3 - x_1)^2} \,, \tag{A.29}$$

we obtain the equation

$$6k^2 N_T^3 C_{S^2} f_{VVV} \mu^{-2} = c \,. \tag{A.30}$$

We have now obtained the three equations (A.21), (A.27) and (A.30), which allow us to compute the function $F(k)$ appearing in the string sphere partition function (2.14). From (A.11), we have

$$F(k) = k^2 N_I^2 C_{S^2} f_{VV} = \frac{k^2 C_{S^2} f_{VV}^3}{f_{VVV}^2} = \frac{c}{6\mu} \,, \tag{A.31}$$

where we used (A.21) in the first equality and the corresponding ratio of (A.27) and (A.30) in the second. This is the expected result as explained after eq. (3.7).

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
