# Peer review of "Holographic Weyl anomaly in string theory"

_SciPost Physics, doi:SciPost Phys. 16, 027 (2024)_

## Round 1 · Referee Report · Anonymous (Referee 1) · 2023-11-8

Strengths

This is a well-written and timely paper.

Weaknesses

None that I can think of.

Report

The authors explain how for non-compact target spacetimes one might extract the sphere amplitude with no insertions. Their main focus is superstring propagation on $AdS_3 \times S^3$, for which they extract the boundary conformal anomaly. The basic logic is well spelt out, and the main calculation is clearly explained in an Appendix.
I think the paper should be accepted for publication in its current form.

---

## Round 1 · Referee Report · Anonymous (Referee 2) · 2023-11-9

Strengths

  1. AdS3-CFT2 is a very important and long studied system that, historically, has provided many insights into the dynamics of quantum gravity. The authors convincingly solve the long-standing question about computing the worldsheet sphere partition function (with no vertex operators) in the AdS3 pure NS background. The result is completely consistent with expectations from the S^2 partition function of the dual CFT2.

  2. This result opens the door to compute the sphere partition function in 3D solid cylindrical geometries, which are discrete orbifolds of the 3D ball geometry studied in this paper. This would allow the computation of the Casimir energy of the dual CFT2 on a circle * time (using the bulk thermal AdS3 background), and the entropy of the BTZ black hole (using the Euclidean BTZ background). It would be interesting future work to compare the BTZ entropy obtained using the method of this paper to the BTZ entropy obtained in the recent work of Halder and Jafferis (arXiv: 2310.02313).

Weaknesses

  1. The technique used to calculate the sphere partition function is somewhat limited with regards to which worldsheet theories it can be applied to. There is a "ghost dilaton operator" which is more universal, and future work could be done on calculating the three point function of this ghost dilaton operator.

  2. Please see the requested changes below.

Report

As desired by expectations of publishing in this journal, this paper presents a breakthrough on a previously-identified and long-standing research stumbling block, namely, the computation of worldsheet sphere partition functions, without any vertex operators, in nontrivial string backgrounds.

This paper meets all of the general acceptance criteria for this journal as well, and therefore should be accepted for publication.

Requested changes

  1. Please check the sign of the linear dilaton R phi term in equations (2.5) and (2.6). On a high genus Riemann surface, R would be negative, and if the path integral is weighted by exp(-S) ~ exp(R phi - e^{2phi}), there would be a divergence from the phi -> -infty end of the integral.

  2. Equation (2.15) contains a term mu log mu. It seems intriguing that in the interpretation, the mu outside the logarithm gets replaced by the inverse string coupling, while the mu inside the logarithm gets replaced by the radius of the sphere on which the dual CFT2 is placed. It would be nice if the authors could discuss this distinction a bit more.

  3. One of the two equations (3.4) and (3.5) seems to have a typo, or more clarification is needed. If (3.4) is indeed the induced metric, then it is mu^{-1} (and not mu) that should be equal to R^2.

---

## Round 1 · Referee Report · Anonymous (Referee 3) · 2023-11-26

Strengths

1-the paper addresses an outstanding problem: the computation of the tree-level partition function of string theory on global AdS_3 with pure NS-NS background, finding results consistent with AdS/CFT expectations.

Weaknesses

-see below under requested changes

Report

This paper solves an outstanding problem in a clear way, and it should be published.

Requested changes

1-The combination of a shift in phi and a corresponding rescaling of gamma is an isometry of AdS_3, and as such the sigma model is (naively) independent of mu. The fact that the partition function does have a mu dependence is a rephrasing of the Weyl anomaly: the symmetry that would remove the mu-dependence is broken quantum mechanically. The author refer to the mu dependence as KPZ scaling, and they provide a brief discussion in section 2, referring in part to earlier literature ([16, 19-21]). As this is an important point (this is the origin of the anomaly), the author should provide more details on the derivation of (2.5), including the Jacobian factors, and why there is non-trivial mu dependence.

A related question is the following: when the CFT contains operators of suitable dimension, the Weyl anomaly receives additional contributions. For example, this is case when the spectrum contains scalar operators of dimension Delta = d/2 + k, where k is an integer, which is common in CFTs appearing in AdS/CFT. Does the KPZ scaling in (2.7) accounts for such additional contributions?

---

## Round 2 · Author Response

Dear Editor,
We would like to thank both referees for their careful reading of the manuscript and particularly referees 2 and 3 for their comments. We have made relevant changes to the manuscript and hereby we are resubmitting the manuscript.
Best,
Lorenz and Sridip
We would like to thank both referees for their careful reading of the manuscript and particularly referees 2 and 3 for their comments. We have made relevant changes to the manuscript and hereby we are resubmitting the manuscript.
Best,
Lorenz and Sridip

---

## Round 2 · List of Changes

- Fixed sign in front of linear dilaton term
- Removed appearances of $\beta$ and $\bar{\beta}$, since they are only a distraction and were partly responsible for the confusion involving $\mu$ and $\mu^{-1}$.
- Redefined $\mu \to \mu^{-1}$
- Added the second referee to the acknowledgments
- Included ways to derive the Jacobian of the path integral measure
- Included a reference to KPZ

---

## Editorial Decision

published